# The Relationship between Professional Variables and Burnout Syndrome in Brazilian Dentists during the COVID-19 Pandemic

**DOI:** 10.3390/ijerph21040435

**Published:** 2024-04-03

**Authors:** Marcelo Salmazo Castro, Gabriela de Figueiredo Meira, Rharessa Gabrielly Ferreira Mendes, Ana Virgínia Santana Sampaio Castilho, Leonardo de Aguiar Trench, Conrado Rodrigues Segalla, Mario Vianna Vettore, Silvia Helena de Carvalho Sales-Peres

**Affiliations:** 1Department of Pediatric Dentistry, Orthodontics and Public Health, Bauru School of Dentistry, University of São Paulo, Bauru 17012-901, SP, Brazil; marcelocastro@usp.br (M.S.C.); gabrielameira@usp.br (G.d.F.M.); rharessa@usp.br (R.G.F.M.); anavcastilho@usp.br (A.V.S.S.C.); leonardotrench@gmail.com (L.d.A.T.); 2Philosophy of Law, Pontifical Catholic University of São Paulo, São Paulo 05015-000, SP, Brazil; segallaadvocacia@gmail.com; 3Department of Dentistry and Oral Health, Aarhus University, Vennelyst Boulevard 9, DK-8000 Aarhus, Denmark; m.vettore@dent.au.dk

**Keywords:** COVID-19, pandemic, burnout, professional, dentists

## Abstract

Burnout syndrome (BS) is a highly prevalent occupational disease among dentists who, during the COVID-19 pandemic, have been at greater risk of contracting the disease, generating stress and distancing. The aim of this study was to assess the association of social conditions, professional factors and perceptions of COVID-19 with Burnout Syndrome. This was a cross-sectional study of 302 Brazilian dentists working in the clinical and private sectors. The professionals completed the Oldenburg Burnout Inventory online and answered sociodemographic and professional questions and questions related to their perception of the pandemic. Poisson regression with unadjusted and adjusted robust variance was used to estimate the association between burnout syndrome (dependent variable) and the independent variables. The presence of BS was strongly associated with age, training in a private institution, professionals who claimed to have sufficient protective knowledge and fear of being contaminated by SARS-CoV-2 during patient care (*p* < 0.05). The findings of this study show that there has been an impact of the COVID-19 pandemic on the occurrence of Burnout syndrome in dentists, especially, those who worked in the public sector and those who were afraid to work with other health professionals.

## 1. Introduction

Burnout syndrome is a multifactorial syndrome consisting of emotional exhaustion, dehumanization and reduced personal fulfillment at work, which occurs as a result of chronic occupational stress [1]. Professionals’ perception of the discrepancy between their efforts and the goals they achieve at work can generate a series of feelings of frustration and interpersonal stress. When exposure to stressors and frustration occurs over a prolonged period, it can trigger the development of burnout syndrome [2].

Due to occupational risks, stress and the need for constant alertness, dentists have been identified in the literature as professionals at high risk of developing burnout syndrome [3,4,5]. These factors contribute to increased feelings of low job satisfaction and a drop in productivity among professionals affected by this syndrome [3,4,5].

Anxiety is a feeling of fear and apprehension, characterized by tension or discomfort derived from the anticipation of danger due to something unknown or strange. Among the degrees of anxiety, mild anxiety is considered natural and promotes preventive and protective behaviors [6]. However, it is necessary to understand that any level of anxiety, no matter how mild, should be treated from the outset as it could be a precursor to a more adverse anxiety disorder in the future. Full-time dedication or worry, coupled with extreme levels of responsibility and professional stress, highlight the need to understand mental conditions related to human behavior in the context of workers’ health. This is in addition to the appropriate diagnosis of stressors that can compromise professional results, as well as the health of dentists [6].

A contaminated working environment is the main factor in the transmission of new virus strains [7]. In a dental environment, where dentists and their equipment are very close to patients and, in most procedures, there is aerosol production due to the use of high rotation, the probability of contamination by microdroplets from an infected patient is high, in addition to the risk of cross-transmission [8]. In the common routine of any dental office, strict biosafety standards are already used due to the fact that several other viruses and bacteria are present in patients’ salivary and bodily fluids. However, due to the high dissemination power characteristic of SARS-CoV-2, some additional measures have become necessary [8].

The massive amount of negative and worrying information about the effects of the pandemic disseminated in the media generated fear throughout the population. However, as health professionals were in more direct contact with potentially infected people, fear and anxiety are the emotions that most commonly afflicted them in their daily routine. Dentists already face pressure and stress at work, and with the addition of the risks of SARS-CoV-2 contamination for themselves, their staff and their patients, during the pandemic, they were fighting a daily battle to maintain their emotional balance. However, to date, there have not been many studies on the mental health of dentists after the COVID-19 pandemic [9]. In addition, professionals have had to continue carrying out dental work, being exposed to the direct risks of contamination.

It is also unclear from the literature whether greater knowledge obtained by dentists can influence increases in pressure and the fear of exposure to pathogens in the oral cavity or whether this added knowledge can generate a factor of concern in the care provided by these professionals. For this reason, knowledge of the mental health conditions of dentists, faced with the challenge of the pandemic, can guide the care to be adopted in daily routines. Thus, the primary objective of this study was to assess the presence of burnout syndrome in dental professionals and to identify whether variables related to social conditions (gender and age), professional factors (academic training institution, graduation rate, degree of clinical performance, active sector and work hours) and those associated with the COVID-19 pandemic were associated with the presence of burnout syndrome among these professionals (Figure 1).

## 2. Materials and Methods

This work followed the STROBE guidelines for cross-sectional studies [10].

### 2.1. Ethical Aspects

The guidelines of the Declaration of Helsinki were adopted for this study [11]. It was initially submitted to and approved by the Human Research Ethics Committee of the Bauru School of Dentistry of the University of São Paulo (CAAE 35177120.9.0000.5417). All participants filled in a Google Forms form, which contained the Informed Consent Form, agreeing to take part in this research.

### 2.2. Sample

A total of 302 dentists took part in the study. The study effect considered was 0.15, the number of predictors in the model was 13 and the value of α = 0.05, so the power of the study was β = 0.998, calculated using the G*Power software in version 3.1.9.4. In September and October 2020, the researchers posted an invitation on their social and professional networks (Facebook^®^, Instagram^®^) containing a link to access the survey questionnaire online.

### 2.3. Eligibility Criteria

The inclusion criteria were being a dentist regularly registered with the Dental Councils and practicing in a public or private dental clinic or practice. Questionnaires with missing data were excluded.

### 2.4. Study Variables

Information was collected on sociodemographic conditions, such as gender; age group; and professional conditions such as place of work, academic training at a public or private institution, time since graduation, weekly working hours, work as a general practitioner or specialist and sector of activity (public, private or both). In the second part of the questionnaire, participants answered about their perceptions of the COVID-19 pandemic. The last part of the form assessed burnout syndrome using the Oldenburg Burnout Inventory (OLBI) according to the theoretical model (Figure 1).

The OLBI questionnaire is an instrument validated by researchers all over the world and is used in two dimensions, assessing exhaustion and professional detachment through 8 questions in each dimension, with 4 positive and 4 negative statements in the two blocks [12] For each item, there are 4 possible answers which are evaluated using a Likert scale ranging from “completely disagree” to “completely agree” [12]. The choice of the OLBI inventory lies in the fact that it is not restricted exclusively to the affective aspects of exhaustion, but rather incorporates questions also related to the physical and cognitive prisms, facilitating the application of the instrument to workers who perform physical work and those whose work is mainly about information processing [12].

In order to evaluate the mean values obtained, due to the fact that the OLBI questionnaire contains questions with negative and positive contexts, it was necessary to invert the values of questions EE3, EE5, EE6, EE8, DT1, DT4, DT6 and DT8 [13] (Table 1).

To characterize the presence or absence of burnout syndrome, the proposal by Peterson et al. [13] was used, where average scores ≥ 2.25 in the exhaustion dimension (referring to questions 1 to 8) and ≥2.1 in the distancing dimension (referring to questions 9 to 16) were considered high, and the participants were classified as shown in Table 2.

The participants were divided into four groups according to the mean scores obtained on the OLBI questionnaire: GSB (without exhaustion and without distancing), GD (with distancing and without exhaustion), GE (with exhaustion and without distancing) and GB (with exhaustion and with distancing) [13] (Figure 2).

### 2.5. Psychometric Analysis of the Oldenburg Burnout Inventory

In order to verify the effectiveness and reliability of the questionnaire in the sample, a confirmatory factor analysis was re-performed by checking the quality using the chi-square test of fit (χ^2^) and the CFI and RMSEA measures, taking into account the criteria adopted by Kline [14]. According to Kline, the viability of the OLBI inventory can be verified by the results obtained through the Cronbach’s alpha (α = 0.916), CFI (0.905) and RMSEA (0.0874) found in this study, with values above 0.90 for both RMSE and CFI being desirable, and RMSE indices close to 0.09 [14,15].

### 2.6. Statistical Analysis

The results found regarding sociodemographic and professional characteristics and those related to the COVID-19 pandemic were evaluated using descriptive analysis. The relationship between the independent variables and the outcome, presence of burnout (dependent variable), was measured using Poisson regression with unadjusted and adjusted robust variance. Variables with *p* ≤ 0.20 were included in the *p* ≤ 0.05 adjusted analysis. The results were presented as Prevalence Ratios (PRs) and their respective 95% CIs (confidence intervals).

## 3. Results

### 3.1. Sociodemographic Analysis

The sample consisted of 302 dentists, 214 of whom were female (70.8%) and 88 male (29.2%). The age groups were divided into 20–30 years 54 (17.9%), 31–50 years 177 (58.6%) and 51 years or more 71 (23.5%). With regard to the undergraduate institution, 185 attended public institutions (61.3%) and 117 attended private institutions (38.7%). As for the sector of work, 199 (66.1%) worked exclusively in public institutions, 46 (15.3%) worked only in the private sector and 56 (18.6%) worked in both sectors. With regard to time since graduation, 50 (16.6%) interviewees had graduated within 15 years, while 120 (39.7%) and 182 (60.3%) had graduated more than 15 years previously.

With regard to clinical practice, 172 (57%) were specialists and 130 (43%) were general practitioners. When asked about their weekly working hours, 66 (21.9%) participants said they worked between 1 and 20 h, 66 (21.9%) between 21 and 40 h, and 143 (47%) and 93 (30%) more than 40 h.

### 3.2. Dentists’ Perception of the COVID-19 Pandemic

With the onset of the pandemic, many dental professionals felt exposed to possible contamination while carrying out dental procedures. Through this questionnaire, it was observed that 226 (74.8%) reported being afraid of contamination during patient care and 281 (93%) of the professionals were postponing appointments for patients with suspicious symptoms. It was also reported that 260 (86.1%) of the interviewees were afraid of becoming infected and transmitting the virus to their family members; 163 (54%) felt afraid when they heard reports of deaths caused by SARS-CoV-2, 213 (70.5%) knew a dentist who had been infected by the virus and 91 (30.1%) of the individuals interviewed had already tested positive for SARS-CoV-2. With regard to the knowledge required for prevention, 267 (88%) said they had the knowledge to protect themselves from contamination by the virus (Table 3).

### 3.3. Analysis of the Burnout Syndrome Questionnaire

Among the main findings, we can note that the majority of participants stated that they feel tired before arriving at work (61%); they need more time to relax than before (68%); that they sometimes feel fed up with their tasks; and that after work they feel low on energy (64%). On the other hand, it was found that the majority also cope very well with the pressures of work (64%); manage their workload well (75%); and often find new and interesting cases (77%). Of the dentists interviewed, 155 (51.5%) presented burnout syndrome, 57 (18.90%) exhaustion, 34 (11.3%) detachment and 55 (18.3%) neither detachment nor exhaustion.

### 3.4. Association between Variables and the Presence of Burnout

The unadjusted analyses of the demographic and professional variables and perceptions related to the COVID-19 pandemic are shown in Table 4. With regard to the age of the professional, dentists aged between 31 and 50 (PR 0.81, 95% CI 0.70–0.93) and those over 50 (PR 0.61, 95% CI 0.47–0.79) were protected against burnout syndrome, and these associations were significant. When the professional variables and the outcome were analyzed, professionals who graduated from private institutions (PR 1.04, 95% CI 0.96–1.09) and who worked in the public sector (PR 1.25, 95% CI 1.04–1.51) and in both the public and private sectors (PR 1.24, 95% CI 1.04–1.48) presented burnout. Dentists with a longer period of training (PR 0.72, 95% CI 0.62–0.83) and those working more than 40 h a week (PR 0.80, 95% CI 0.64–1.00) were less likely to develop the syndrome.

Burnout was associated with the fear of becoming infected during professional practice (PR 1.49, 95% CI 1.15–1.93), knowing a dentist who had already been infected with SARS-CoV-2 (PR 1.29, 95% CI 1.11–1.49), the fear of transmitting the virus to their family members (PR 1. 74, 95% CI 1.15–2.62), those who reported having protective knowledge against contamination (PR 0.76, 95% CI 0.66–0.88), as well as dentists who reported being afraid to work with other health professionals (PR 1.44, 95% CI 1.23–1.68).

In the adjusted model, dentists who worked in the public sector (PR 1.24, 95% CI 1.02–1.50) and who reported a fear of working with other health professionals (PR 1.27, 95% CI 1.09–1.40) had a higher prevalence of burnout syndrome. The results found in the adjusted and unadjusted analyses within the study’s theoretical model are shown in Figure 3.

## 4. Discussion

The results of this study add a new perspective to the literature and the reality of dental practice, since the COVID-19 pandemic has brought a new scenario to the job market and to health research. Recent studies have shown that the quality of life of dental professionals has been more greatly impacted, such as through a fear of becoming infected and dying from SARS-CoV-2 [16,17]. However, our study adds the perception of the pandemic in the presence of burnout syndrome, considered a limiting syndrome with a high prevalence among health professionals, and Brazil was considered the country with the second highest number of affected professionals according to the WHO [18].

Our results showed that dentists between the ages of 20 and 30 were more exposed to burnout. This relationship was also demonstrated by Antoniadou (2022) [19], who found that dentists between the ages of 25 and 35 showed greater burnout during the COVID-19 pandemic, mainly due to the challenges of the job market [19]. In our sample, the lower exposure in older dentists can be explained by the time of data collection, which corresponded to the first wave of the pandemic, when Brazil had a high prevalence of COVID-19, especially in older people, and at that time, there were still no vaccines, so the fear of dying from the infection in the workplace generated anguish and anxiety, thus over-riding financial factors, leading these professionals to cancel appointments [16].

Our results also show that professionals who worked as a dentist in public institutions were more likely to suffer from burnout. This relationship can be explained by the fact that many public sector professionals during the pandemic were reallocated to other activities within the health sector, such as collecting SARS-CoV-2 tests. This was statistically proven in our results, reinforcing the fact that dentists who did not have sufficient protective knowledge showed burnout. This finding was worrying, as these professionals routinely perform procedures with instruments that generate aerosols, which can be inhaled in the workplace, exposing both the clinician and the patient to infection.

Dentists must protect patients from the spread of all infectious diseases by adopting safe practices in the performance of their work and making dental procedures risk-free [9]. It should also be noted that new biosafety protocols have been created and adjusted according to the evolution of scientific knowledge about the risks of infection by SARS-CoV-2 [19], generating doubts and impacting the work of health professionals [20].

Our data are in line with a study carried out in Turkey by Özarslan, Caliskan (2020); the authors reported that the level of exhaustion and stress among dentists was higher in those who followed the protocols recommended by the WHO, such as the use of an N95 or FFP3 mask, a surgical mask, goggles, a face shield and a cap for aerosol-generating care. Therefore, protective knowledge alone was not enough to prevent stress among dental professionals in many parts of the world [21]. In this study, most of the participants showed burnout during the COVID-19 pandemic. It is worth mentioning that in Brazil, dentists working in the public sector were called to work on the front line, performing non-routine procedures such as screening patients with symptoms, administering vaccines and testing patients with suspected acute respiratory syndrome [16]. These data corroborate scientific evidence which suggests that health professionals who work on the front line are more prone to mental disorders such as anxiety, depression, post-traumatic stress and sleep disorders, which has a greater impact on their quality of life [22].

Our findings have some limitations: surveys using forms are more prone to information bias, as participants may consult a source while answering the questionnaires. Another limitation is that the sample was selected randomly because the participants were in the researchers’ social network. However, professionals from all over Brazil took part, which minimizes this initial effect. However, this type of research opened up opportunities, given that we were in a period of social distancing and isolation. The size of the sample can also be a limiting factor in extrapolating the results, which was minimized by the greater power of the study.

However, to our knowledge, few studies have been published on the relationship between burnout and dental professionals during the pandemic, mainly in Asian and European countries. It is worth noting that in these studies, professional or sociodemographic variables were associated with burnout and professional distancing, while the present study addresses the professional and personal perspective in the face of this occupational disease. Our results showed that the fear of infection at the time of professional activity outweighed all other factors, even in professionals who had protective knowledge of pandemic situations. Furthermore, the present study contemplates the need for public policies to be created with the aim of preventing, as well as contemplating, the monitoring and treatment of occupational diseases with an emotional scope such as burnout syndrome. This occupational disease is compounded by factors associated with the global context, making it necessary to look far beyond the dental chair and the presence of the patient in the dental environment, and these effects may still be with us even though the worst of the pandemic’s clinical impact is over.

## 5. Conclusions

The results of this study show that dentists who working in the public sector and who were afraid to work with other health professionals, had Burnout syndrome. It is necessary for dental professionals to be psychologically and physically prepared to deal with adverse situations and for preventive measures to be employed to improve the quality of life and professional performance of dentists.

## Figures and Tables

**Figure 1 ijerph-21-00435-f001:**
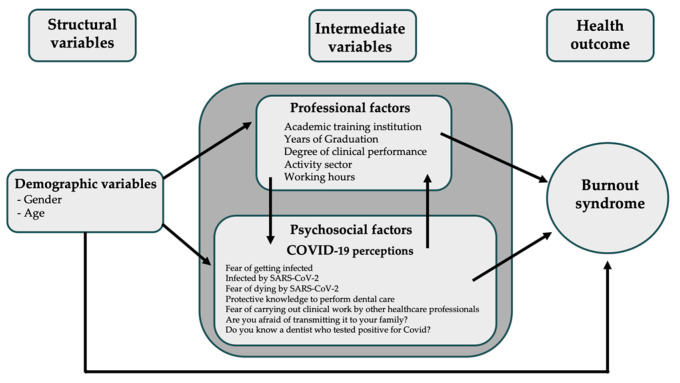
Theoretical model of the influence of study variables and the presence of burnout syndrome in dental professionals.

**Figure 2 ijerph-21-00435-f002:**
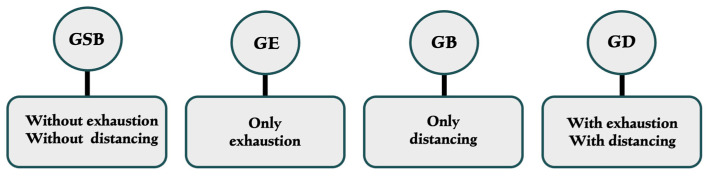
Characterization of study groups—SB (group without exhaustion and distancing), GE (group with exhaustion), GD (group with distancing) and GB (group with exhaustion and distancing).

**Figure 3 ijerph-21-00435-f003:**
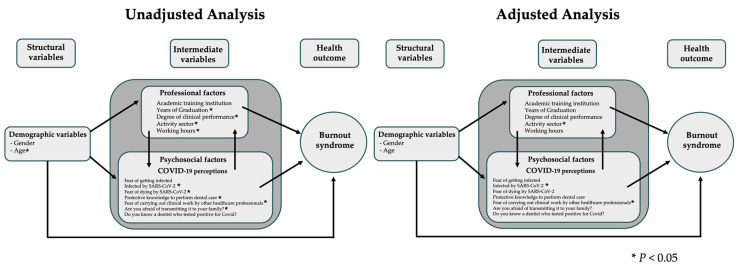
Unadjusted and adjusted Poisson regression in the theoretical model.

**Table 1 ijerph-21-00435-t001:** Oldenburg Burnout Inventory.

Questions	Number
There are days when I feel tired before I even get to work.	EE1
After work, I need more time to feel better than I used to.	EE2
I can handle the pressures of my job very well.	EE3
During my job, I feel emotionally exhausted.	EE4
After professional tasks, I have energy for my leisure activities.	EE5
When I work, I feel good.	EE6
After work I feel tired and without energy.	EE7
In general, I can manage the amount of work I have well.	EE8
I often do new and interesting things in my job.	DT1
I increasingly speak more and more often in a negative way about my job.	DT2
Lately, I have been carrying out my work almost mechanically.	DT3
I consider my job a positive challenge.	DT4
As time passes, I become disinterested in my job.	DT5
The work I do today is the only one I imagine myself doing.	DT6
I feel more and more committed to my work.	DT7
I often feel fed up with my tasks.	DT8

**Table 2 ijerph-21-00435-t002:** Interpretive classification of the presence or absence of burnout syndrome.

Dimension		
Exhaustion	Distancing	Classification
Low	Low	Without exhaustion and distancing
Low	High	With distancing
High	Low	With exhaustion
High	High	With Burnout

**Table 3 ijerph-21-00435-t003:** Dentists’ perceptions of the COVID-19 pandemic.

Dentists’ Perceptions	Yes *n* (%)	No *n* (%)
Are you afraid of being infected with COVID-19 while carrying out a dental procedure?	226 (75)	76 (25)
Are you afraid of becoming infected during your work and transmitting the virus to your family?	260 (86)	42 (14)
Have you ever tested positive for COVID-19?	91 (30)	211 (70)
Do you feel scared when you hear that people are dying from SARS-CoV-2?	258 (84)	44 (16)
Do you think you have the necessary protective knowledge regarding COVID-19?	267 (88)	35 (12)

**Table 4 ijerph-21-00435-t004:** Unadjusted and adjusted Poisson regression between sociodemographic, professional and COVID-19 pandemic-related variables with the presence of burnout in Brazilian dentists.

	Unadjusted Analysis			Adjusted Analysis		
Demographic variables						
	**PR**	**IC 95%**	** *p* **	**PR**	**IC 95%**	** *p* **
Gender						
Male	1					
Female	1.11	0.92–1.36	0.25			
Age						
20 to 30 years	1					
31 to 50 years	0.81	0.70–0.93	<0.04 *	0.99	0.82–1.19	0.95
>50 years	0.61	0.47–0.78	<0.01 *	0.77	0.56–1.05	0.10
Professional variables						
Academic training institution						
Public	1					
Private	1.04	0.87–1.23	0.63			
Years since Graduation						
0 to 15 years	1					
More than 15 years	0.72	0.62–0.83	<0.01 *	0.86	0.71–1.05	0.16
Degree of clinical performance						
Specialist dentist	1			1		
General dentist	0.83	0.71–0.97	<0.05 *	0.90	0.78–1.04	0.17
Activity sector						
Private	1			1		
Public	1.25	1.04–1.51	<0.05 *	1.24	1.02–1.50	<0.05 *
Both	1.24	1.04–1.48	0.05 *	1.22	0.99–1.51	0.06
Working Hours						
1 to20 h	1			1		
21 to 40 h	0.91	0.61–1.10	0.36	0.94	0.80–1.12	0.53
+40 h	0.80	0.64–1.00	0.05 *	0.85	0.68–1.07	0.17
COVID-19 Perceptions						
Fear of being infected while caring for a patient						
No	1			1		
Yes	1.49	1.15–1.93	<0.02 *	1.25	0.93–1.68	0.13
Infected by SARS-CoV-2						
No	1			1		
Yes	1.29	1.11–1.49	<0.01 *	1.15	0.98–1.34	0.05 *
Fear of dying by COVID-19						
No	1			1		
Yes	1.20	0.91–1.59	0.19	0.76	0.54–1.05	0.09
Protective knowledge to perform dental care						
No	0.76	0.66–0.88	<0.01 *	0.96	0.80–1.16	0.70
Yes	1			1		
Fear of carrying out clinical work with other healthcare professionals						
No	1			1		
Yes	1.44	1.23–1.68	<0.01 *	1.27	1.09–1.40	<0.05 *
Are you afraid of transmitting it to your family?						
No	1			1		
Yes	1.74	1.15–2.62	<0.01 *	1.26	0.80–1.98	0.31
Do you know a dentist who tested positive for COVID?						
No	1					
Yes	1.06	0.87–1.29	0.52			

* *p* < 0.05.

## Data Availability

The data presented in this study are available on request from the corresponding author.

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
