# Peer review of "The Relationship between Professional Variables and Burnout Syndrome in Brazilian Dentists during the COVID-19 Pandemic"

_ijerph, 2024, doi:10.3390/ijerph21040435_

Round 1
Reviewer 1 Report
Comments and Suggestions for Authors
Thank you for the opportunity to review this manuscript. There are a few issues with this manuscript that need to be addressed.
Introduction
Page 2, lines 47-50: You introduce the concept of various degrees of anxiety but you only mention "mild anxiety". It would have been good to introduce what different degrees of anxiety look like including physiological, cognitive and emotional characteristics. Also, the assertion that all "mild" anxiety is productive should be challenged as any form of anxiety can be a precursor of a more adverse anxiety disorder in the future.
Page 2, lines 50-51: your statement "Full-time dedication or worry, combined with extreme responsibil-50 ity and professional stress, highlight the need to understand mental conditions re-51 lated to human behavior in the context of workers' health" t this could be an interesting hypothesis to explore as part of this work rather than a statement in your introduction as it it unsubstantiated and feels out of place.
Page 2, lines 52-54: It is not clear why the diagnosis of stressors will impact patient safety. This statement feels stigmatising of dental professionals who are diagnosed with anxiety or other mental health disorders as it they are unable to perform their work properly even if their issues are well managed. Please consider a more nuanced approach in your statements.
Page 3, figure 1: please amend the title as it appears in a mixture of English and Portuguese.
Page 3, line 105: Can you explain why this approach was used in measuring COVID-19 perceptions? Did you consider any other questionnaires before settling in developing your own version? How did you settle on these specific questions? Some additional information is necessary to support the use of that particular approach.
Results: I would have expected to see figure 1 repeated in your results with the outcomes of your analyses to confirm or reject your proposed model. Please consider a better way of representing your outcomes to make it clear how you examined the theoretical model you proposed earlier on.
Page 9, lines 243-244 and page 9, lines 258-259: these are two contradicting statements. On one hand (lines 243-244) you state that those from private institutions lacked protective knowledge and therefore were more exposed to burnout. Then (lines 258-259) you say your results are in line with a study in Turkey where participants were less protected if they showed protective knowledge that contradicts your previous statement. Is protective knowledge enough to prevent burnout or not?
Discussion: a mention on clinical, research and policy implications of this work should have been included.
Comments on the Quality of English Language
Minor English language editing is needed.
Author Response
Author’s response:
Question 1 - Page 2, lines 47-50: You introduce the concept of various degrees of anxiety but you only mention "mild anxiety". It would have been good to introduce what different degrees of anxiety look like including physiological, cognitive and emotional characteristics. Also, the assertion that all "mild" anxiety is productive should be challenged as any form of anxiety can be a precursor of a more adverse anxiety disorder in the future.
The correction has been added to the text.
Page 2, lines 50-51: your statement "Full-time dedication or worry, combined with extreme responsibil-50 ity and professional stress, highlight the need to understand mental conditions re-51 lated to human behavior in the context of workers' health" t this could be an interesting hypothesis to explore as part of this work rather than a statement in your introduction as it it unsubstantiated and feels out of place.
This statement was to contextualize the importance of this study to help understand the variables associated with the presence of Burnout syndrome.
Page 2, lines 52-54: It is not clear why the diagnosis of stressors will impact patient safety. This statement feels stigmatising of dental professionals who are diagnosed with anxiety or other mental health disorders as it they are unable to perform their work properly even if their issues are well managed. Please consider a more nuanced approach in your statements.
This statement was removed from the text.
Page 3, figure 1: please amend the title as it appears in a mixture of English and Portuguese.
The title has been corrected
Page 3, line 105: Can you explain why this approach was used in measuring COVID-19 perceptions? Did you consider any other questionnaires before settling in developing your own version? How did you settle on these specific questions? Some additional information is necessary to support the use of that particular approach.
The questionnaire was developed based on parameters from other studies collected by a systematic review carried out by our working group and the reference has already been inserted in the text.
Results: I would have expected to see figure 1 repeated in your results with the outcomes of your analyses to confirm or reject your proposed model. Please consider a better way of representing your outcomes to make it clear how you examined the theoretical model you proposed earlier on.
The results have been added in a figure 2
Page 9, lines 243-244 and page 9, lines 258-259: these are two contradicting statements. On one hand (lines 243-244) you state that those from private institutions lacked protective knowledge and therefore were more exposed to burnout. Then (lines 258-259) you say your results are in line with a study in Turkey where participants were less protected if they showed protective knowledge that contradicts your previous statement. Is protective knowledge enough to prevent burnout or not?
The statement has been corrected
Discussion: a mention on clinical, research and policy implications of this work should have been included.
The mention has been added to the text
Reviewer 2 Report
Comments and Suggestions for Authors
The article, “Burnout among dental professionals during COVID-19 pandemic,” investigates a topic that has important implications for the well-being of individual dentists and for the future of the field of dentistry. The article is generally well-written and the data were appropriately collected and analyzed. The manuscript, however, would have more impact if the authors considered a few enhancements.
First, the title implies that this is a descriptive study of the prevalence of burnout during the pandemic, but the data analysis goes significantly beyond description by identifying independent and dependent variables and by testing a hypothesis. The authors should consider a title that more accurately reflects the analysis by indicating the relationship between variables. For example: “Dental professionals’ characteristics associated with burnout during the COVID-19 pandemic”.
The Introduction is well researched and includes useful information. It narrows to the primary objective in the Introduction’s final sentence (p. 2, lines 78-81), which mentions “variables associated with Burnout Syndrome in these professionals.” The authors should provide a brief elaboration of those variables and why they were selected. Without that justification for hypothesis testing, only when readers arrive at the study variables subsection (p. 3, lines 101-108) do they learn what those variables include. The authors never explain, for instance, why they believe public vs. private institution or generalist vs. specialist might be associated with burnout during the Covid-19 pandemic. Without that explanation, it appears that data were collected and regressed on a burnout measure to see if anything was significant (and publishable). Given that it identifies the variables, perhaps Figure 1 would be better placed in the Introduction instead of the Materials and Methods?
Also, reorganization of the Introduction might enhance the relevance and impact of the article. The first three paragraphs explore the pandemic and its associated anxiety. Not until the end of the fourth paragraph (p. 2, line 58) is burnout syndrome mentioned. Then the authors go on to detail the problem of burnout among dentists—a problem which predated the pandemic and which presumably persists now that the worst of the pandemic is over. Readers therefore may be less interested in the problem of burnout during a pandemic which is effectively over, than in the ways Covid-19 made a chronic problem worse—perhaps to the present day. This is simply a matter of framing the rationale for the study. Starting with the description of the significant and ongoing problem of dentist burnout is likely to appeal to more readers than starting with yet another description of how bad the pandemic was four years ago.
In the Materials and Methods section, the authors state (p. 2, line 83), “This work followed the STROBE guidelines for cross-sectional studies [11].” Yet citation 11 is the Demerouti et al. article comparing the OLdenburg Burnout Inventory (OLBI) and the Maslach Burnout Inventory – General Survey (MBI-GS). That article never mentions the STROBE guidelines. Further, the full reference (p. 11, line 320) indicates it was published in the year “3003” rather than in 2003.
In the Results section, Table 4 (p. 6, lines 186+) is unnecessary and is not formatted correctly. It is unnecessary because few, if any, readers will be interested in knowing the frequencies for every item in the OLBI questionnaire. The summary of the noteworthy frequencies (p. 7, lines 187-192) and your summary of the scores on the two factors (p. 7, lines 192-194) is sufficient. However, that summary of the four categories ends with a reference to Table 3 (p. 7, line 194), but Table 3 does not indicate the number or percent of dentists in each of the four categories, so the reference should be eliminated. Table 3 is formatted incorrectly because one row (“I always find new and interesting aspects in my work”) is not aligned correctly. Also, some of the percentages include no decimal points others include a decimal to the tenth of a percentage. And some of the decimals use commas whereas others use periods. Finally, the first column, with the measure’s items written out, is narrow and difficult to read.
The presentation of older age as a protective factor (p. 7, line 199) is somewhat confusing given that all of the other data are presented in terms of risk factors.
In the Discussion section, the authors should revise the sentence (p. 9, lines 224-226): “Recent studies have shown that dental professionals have had a greater impact on quality of life, such as fear of becoming infected and dying from SARS-CoV-2 [15,16].” It is unclear what variables are being described. How do dental professionals have an impact on quality of life? Whose quality of life?
A citation is missing on p. 9, line 231.
The explanation why older dentists experience less burnout than younger dentists (p. 9, lines 233-239) is very confusing. The authors note that older people were more anxious and vulnerable to infection and death during the months of data collection, but then indicate this is a reason they experienced less burnout. Do the authors mean the older dentists were less likely to work during those months and therefore were less susceptible to burnout? If so, they should clarify the point.
The explanation of the relationship between private institution training and lack of biosafety knowledge (p. 9, lines 240-247) should be clarified, as well. In general, new data and analysis should not be presented in the Discussion section. Any additional statistical tests demonstrating a correlation between private training and biosafety knowledge should be added to the appropriate subsection in the Results. Then the authors could elaborate on that point in the Discussion. I was also left wondering why public institutions address infection control more than private ones do?
Brackets indicating a citation are missing on p. 10, line 264.
The paragraph acknowledging limitations (p. 10. Lines 268-272) should also acknowledge a potential selection effect in the sampling method. Specifically, all of the participants were presumably in the same social network (all known to the researchers) and therefore may be similar in ways that distinguish them from the larger population of Brazilian dentists.
Page 10, line 283 references “the owl” and I do not understand what that reference means. That paragraph also notes that “pandemics are situations that can happen at any time.” While this is a true statement, the larger impact of your research might be that this specific pandemic may have amplified existing burnout, especially among dentists with certain characteristics, and those effects may still be with us even though the worst of the pandemic’s clinical impact is over.
Regarding the writing, different publications have different guidance regarding the capitalization of Covid-19, but the use should be consistent throughout. At present, the title (p. 1, line 2) includes all capitals (COVID-19) and the first sentence of the abstract (p. 1, line 16) includes only the initial letter capitalized (Covid-19). The editor can inform you about the proper capitalization of that term in the International Journal of Environmental Research and Public Health.
The authors alternate between the term “dentists” (p. 2, line 62 and elsewhere) and “dental surgeons” (p. 2, line 69 and elsewhere). In some nations, including the United States, a dental surgeon has significantly more training than a general dentist and their work is quite distinct. The authors also distinguish between general dentists and specialists in their data collection. To avoid confusion, the authors should consistently use one term—preferably “dentist” because that term includes dental surgeons, as well. Also, Table 4 (p. 8) mentions “Especialist dentist” when the term in English should be “Specialist dentist.” Similarly, other references in the first column include Portuguese terms (e.g. in the categories “Age group,” “Graduation rate,” and “Work Hours”).
Also, the authors should avoid gendered language which presumes dentists are male by referring to “his work” and “his patients” (p. 2, lines 68-69). An easy way to avoid this problem is using the plural form—as the authors did when saying “they must be informed” (p. 2, line 70).
The title of Figure 1 (p. 3, lines 110-111) includes both English and (I assume) Portuguese. Was this intentional?
A few lines lower (p. 3, line 115) the future tense is used: “…4 possible answers which will be….” The past tense should be used in this article (e.g. “…4 possible answers were…”). Similarly, on p. 3, line 123, “…it is necessary to…” should read, “…it was necessary to….” Present and future tense is also used elsewhere when past tense should be used (e.g. p. 4, line 130; p. 9, line 237).
Text is repeated at the beginning of the sentence on p. 3, line 117: “The choice of the OLBI inventory, the choice of the OLBI inventory….”
I congratulate the authors on a worthwhile study and hope that it will eventually result in publication.
Comments on the Quality of English LanguageSee comments above. English is good overall, but some minor editing is needed to adjust verb tense in a few instances and to clarify the meaning of a few sentences. Also, in a few places Portuguese is used in place of, or alongside, the English terms.
Author Response
Author’s response:
Question1 – The article, “Burnout among dental professionals during COVID-19 pandemic,” investigates a topic that has important implications for the well-being of individual dentists and for the future of the field of dentistry. The article is generally well-written and the data were appropriately collected and analyzed. The manuscript, however, would have more impact if the authors considered a few enhancements.
First, the title implies that this is a descriptive study of the prevalence of burnout during the pandemic, but the data analysis goes significantly beyond description by identifying independent and dependent variables and by testing a hypothesis. The authors should consider a title that more accurately reflects the analysis by indicating the relationship between variables. For example: “Dental professionals’ characteristics associated with burnout during the COVID-19 pandemic”.
According to your observation, the authors have changed the title to “Impact of professional variables on burnout syndrome in Brazilian dentists during the COVID-19 pandemic”
Question 2- The Introduction is well researched and includes useful information. It narrows to the primary objective in the Introduction’s final sentence (p. 2, lines 78-81), which mentions “variables associated with Burnout Syndrome in these professionals.” The authors should provide a brief elaboration of those variables and why they were selected. Without that justification for hypothesis testing, only when readers arrive at the study variables subsection (p. 3, lines 101-108) do they learn what those variables include. The authors never explain, for instance, why they believe public vs. private institution or generalist vs. specialist might be associated with burnout during the Covid-19 pandemic. Without that explanation, it appears that data were collected and regressed on a burnout measure to see if anything was significant (and publishable). Given that it identifies the variables, perhaps Figure 1 would be better placed in the Introduction instead of the Materials and Methods?
The figure of the theoretical model has been relocated to the introduction to improve understanding of the choice of study variables. In addition, we added to the introduction a question about whether or not dentists' greater knowledge could affect their fear of COVID-19 contamination and consequently the appearance of Burnout Syndrome.
Question 3- Also, reorganization of the Introduction might enhance the relevance and impact of the article. The first three paragraphs explore the pandemic and its associated anxiety. Not until the end of the fourth paragraph (p. 2, line 58) is burnout syndrome mentioned. Then the authors go on to detail the problem of burnout among dentists—a problem which predated the pandemic and which presumably persists now that the worst of the pandemic is over. Readers therefore may be less interested in the problem of burnout during a pandemic which is effectively over, than in the ways Covid-19 made a chronic problem worse—perhaps to the present day. This is simply a matter of framing the rationale for the study. Starting with the description of the significant and ongoing problem of dentist burnout is likely to appeal to more readers than starting with yet another description of how bad the pandemic was four years ago.
The introduction was reorganized, prioritizing the topic of Burnout Syndrome
Question 4- In the Materials and Methods section, the authors state (p. 2, line 83), “This work followed the STROBE guidelines for cross-sectional studies [11].” Yet citation 11 is the Demerouti et al. article comparing the OLdenburg Burnout Inventory (OLBI) and the Maslach Burnout Inventory – General Survey (MBI-GS). That article never mentions the STROBE guidelines. Further, the full reference (p. 11, line 320) indicates it was published in the year “3003” rather than in 2003.
The reference for STROBE guidelines has been inserted and the typo for the year "3003" has been corrected
Question 5- In the Results section, Table 4 (p. 6, lines 186+) is unnecessary and is not formatted correctly. It is unnecessary because few, if any, readers will be interested in knowing the frequencies for every item in the OLBI questionnaire. The summary of the noteworthy frequencies (p. 7, lines 187-192) and your summary of the scores on the two factors (p. 7, lines 192-194) is sufficient. However, that summary of the four categories ends with a reference to Table 3 (p. 7, line 194), but Table 3 does not indicate the number or percent of dentists in each of the four categories, so the reference should be eliminated. Table 3 is formatted incorrectly because one row (“I always find new and interesting aspects in my work”) is not aligned correctly. Also, some of the percentages include no decimal points others include a decimal to the tenth of a percentage. And some of the decimals use commas whereas others use periods. Finally, the first column, with the measure’s items written out, is narrow and difficult to read.
The table has been removed
Question 6 - The presentation of older age as a protective factor (p. 7, line 199) is somewhat confusing given that all of the other data are presented in terms of risk factors.
The protective factor for older dentists seems to be correlated with the fact that these professionals are more economically stabilized than younger dentists and therefore, during the peaks of COVID-19 contamination, they were able to reduce or cancel appointments and consequently be less exposed to contamination and professional stress.
Question 7- In the Discussion section, the authors should revise the sentence (p. 9, lines 224-226): “Recent studies have shown that dental professionals have had a greater impact on quality of life, such as fear of becoming infected and dying from SARS-CoV-2 [15,16].” It is unclear what variables are being described. How do dental professionals have an impact on quality of life? Whose quality of life?
The worsening quality of life we are referring to is that of the dentists who attended during the peaks of COVID-19, since these professionals often stopped working for fear of exposure during appointments, reducing their fees, or when they needed to work they were afraid of being contaminated or even of being the agents who brought the contamination to their families.
Question 8- A citation is missing on p. 9, line 231.
The citation has been added
Question 9 -The explanation why older dentists experience less burnout than younger dentists (p. 9, lines 233-239) is very confusing. The authors note that older people were more anxious and vulnerable to infection and death during the months of data collection, but then indicate this is a reason they experienced less burnout. Do the authors mean the older dentists were less likely to work during those months and therefore were less susceptible to burnout? If so, they should clarify the point.
This question was answered in question 6
Question 10 - The explanation of the relationship between private institution training and lack of biosafety knowledge (p. 9, lines 240-247) should be clarified, as well. In general, new data and analysis should not be presented in the Discussion section. Any additional statistical tests demonstrating a correlation between private training and biosafety knowledge should be added to the appropriate subsection in the Results. Then the authors could elaborate on that point in the Discussion. I was also left wondering why public institutions address infection control more than private ones do?
The divergence between the curriculum of public and private training institutions may be one of the hypotheses that explain the difference between the results found.
Question 11- Brackets indicating a citation are missing on p. 10, line 264.
The brackets has been added
Question 12- The paragraph acknowledging limitations (p. 10. Lines 268-272) should also acknowledge a potential selection effect in the sampling method. Specifically, all of the participants were presumably in the same social network (all known to the researchers) and therefore may be similar in ways that distinguish them from the larger population of Brazilian dentists.
This new limitation was inserted into the study
Question 13- Page 10, line 283 references “the owl” and I do not understand what that reference means.
“the owl” it has been changed for the correct term
Question 14- That paragraph also notes that “pandemics are situations that can happen at any time.” While this is a true statement, the larger impact of your research might be that this specific pandemic may have amplified existing burnout, especially among dentists with certain characteristics, and those effects may still be with us even though the worst of the pandemic’s clinical impact is over.
The suggested changes have been added to the text
Regarding the writing, different publications have different guidance regarding the capitalization of Covid-19, but the use should be consistent throughout. At present, the title (p. 1, line 2) includes all capitals (COVID-19) and the first sentence of the abstract (p. 1, line 16) includes only the initial letter capitalized (Covid-19). The editor can inform you about the proper capitalization of that term in the International Journal of Environmental Research and Public Health.
The terms have been standardized
The authors alternate between the term “dentists” (p. 2, line 62 and elsewhere) and “dental surgeons” (p. 2, line 69 and elsewhere). In some nations, including the United States, a dental surgeon has significantly more training than a general dentist and their work is quite distinct. The authors also distinguish between general dentists and specialists in their data collection. To avoid confusion, the authors should consistently use one term—preferably “dentist” because that term includes dental surgeons, as well. Also, Table 4 (p. 8) mentions “Especialist dentist” when the term in English should be “Specialist dentist.” Similarly, other references in the first column include Portuguese terms (e.g. in the categories “Age group,” “Graduation rate,” and “Work Hours”).
The terms have been standardized
Also, the authors should avoid gendered language which presumes dentists are male by referring to “his work” and “his patients” (p. 2, lines 68-69). An easy way to avoid this problem is using the plural form—as the authors did when saying “they must be informed” (p. 2, line 70).
The terms have been standardized
The title of Figure 1 (p. 3, lines 110-111) includes both English and (I assume) Portuguese. Was this intentional?
The title has been corrected
A few lines lower (p. 3, line 115) the future tense is used: “…4 possible answers which will be….” The past tense should be used in this article (e.g. “…4 possible answers were…”). Similarly, on p. 3, line 123, “…it is necessary to…” should read, “…it was necessary to….” Present and future tense is also used elsewhere when past tense should be used (e.g. p. 4, line 130; p. 9, line 237).
The terms has been changed
Text is repeated at the beginning of the sentence on p. 3, line 117: “The choice of the OLBI inventory, the choice of the OLBI inventory….”
The repeated sentence has been removed
Round 2
Reviewer 1 Report
Comments and Suggestions for Authors
Thank you for addressing my comments and for the changes you have made following my initial review. I am happy with the current version of the manuscript and have no further comments.